# Method for Detecting Abnormal Activity in a Group of Mobile Robots

**DOI:** 10.3390/s19184007

**Published:** 2019-09-17

**Authors:** Elena Basan, Alexandr Basan, Alexey Nekrasov

**Affiliations:** 1Department of Information Security, Southern Federal University, 347922 Taganrog, Russia; tftrtu@mail.ru (A.B.); alexei-nekrassov@mail.ru (A.N.); 2LLC “Engineering Center” Integra, 347922 Taganrog, Russia; 3Department of Radio Engineering Systems, Saint Petersburg Electrotechnical University, Professora Popova 5, 197376 Saint Petersburg, Russia; 4Higher Technical School of Computer Engineering, University of Malaga, 29071 Malaga, Spain

**Keywords:** attack, mobile robot, mathematical statistic, anomalies, internal intruder

## Abstract

The range of attacks implemented on wireless networks is quite wide. To avoid or reduce the likelihood of an attack, it is necessary to use various defense mechanisms. Existing protection mechanisms are not always suitable for robotic systems and may not fully provide the necessary level of security. Thus, it is necessary to develop new ways of protection, which would be specific to groups of mobile robots. In this study, we propose an analysis of the following cyber parameters: the power consumption and the residual energy, as well as an in-depth traffic analysis to evaluate the effectiveness of the attack and identify abnormal network. We realized an analysis of the behavior of robotic systems under normal conditions and determined that, by their nature, robotic systems have a static and uniform behavior. We developed an experimental stand, and also conducted a theoretical analysis to confirm our assumptions. We found that some indicators of the components of the robotic system change statically; that is, there was a little deviation from the mean. Thus, we identified a set of metrics that allow us to determine how static the operation of the robotic system and its components is. Metrics allow us to evaluate parameters such as power consumption, and incoming/outgoing/redirected/dropped network packets. The results obtained are important for creating an integrated system for detecting anomalies in robotic systems. At the same time, the robotic node can analyze these parameters independently and make calculations that do not greatly affect performance. The main idea of this paper is to define a set of metrics characterizing the static behavior of a robotic system for the further development of an anomaly detection system.

## 1. Introduction

Today, the popularity of mobile robotic systems is growing. The areas of their application vary from everyday use to the military industry. Mobile robots can be divided into the following four groups: autonomous land vehicles, unmanned aerial vehicles (UAVs), autonomous marine vehicles, and autonomous underwater vehicles (APA) [1]. At the same time, these devices operate quite unstably, and frequent failures are observed. Devices easily fail and can be destroyed. Environmental conditions strongly affect the behavior of the mobile robots. UAVs are especially affected by the environment. Mobile robots are generally controlled by an operator with the use of a wireless channel. Mobile robots are often located outside the controlled area; they have limited computing power and a small-capacity battery [2]. 

There are a large number of attacks aimed at taking control of the mobile robot, and denial of system operation. Such attacks could be implemented at the data link and physical layer by an external attacker [3]. These attacks can cause significant damage to the network. Their goals are the destruction of the system, disruption of the network, and penetration into the mobile robotic system (MRS). If the attacker’s task is to penetrate the MRS, get access to all devices, change the logic of the system operation, and receive confidential information, then the intruder could implement internal active attacks on the MRS. In order to develop an effective system for detecting and blocking network attacks, it is necessary to understand which components of the system are most exposed to network attacks, as well as which indicators the attack affects.

There are several ways to control the MRS: remote control, independent control, overall control, and group control. At the same time, group management systems can be fully distributed or hierarchical, where the group leader stands out. Different technologies are used to organize each type of control system. Wireless communication channels and communication protocols are used to organize the management process of the MRS. In particular, communication protocols are used to transmit control commands, exchange collected data, etc. Table 1 represents management types of mobile robotic systems and features of the wireless network implemented to organize the management process. When identifying anomalous behavior in the groups of mobile robots, it is necessary to clearly understand what technologies and algorithms are used in their development and implementation on the real objects. This is necessary because the structure, algorithms, and methods of functioning mobile systems differ significantly from standard computer networks.

At the same time, the groups of mobile robots are divided into the following types of mobile networks, depending on the functions they perform and the type of mobile robot used:VANET (vehicular network ad hoc network) is the wireless network with (unmanned) vehicle subscribers that implement fully decentralized control in the absence of base stations or reference nodes. Topology rapidly changes with a random connection of nodes.MANET (mobile ad hoc network) is a wireless decentralized network [4] with random mobile devices that is implemented in the absence of base stations or reference nodes. The MANET topology is dynamic, and the connection of nodes is random.FANET (flying ad hoc network), by analogy with the mobile peer-to-peer network MANET and vehicle peer-to-peer network VANET, is a special type of peer-to-peer self-organizing network based on unmanned aerial vehicles (UAVs) [5].

Various protocols and standards can be used for the organization of these types of networks as presented in Table 2.

It can be noted from Table 2 that the developers pay more attention to routing protocols. Indeed, routing in the networks of mobile robots is quite a difficult task, because it is necessary to deliver packets between any two nodes of the network, taking into account changing conditions. 

There were many methods and algorithms developed for organization of the group control system to manage the mobile robot. Overviews of the algorithms and methods of mobile robot control systems are presented in References [6,7]. Despite the variety of methods and approaches to organizing group control of mobile robots, the main stages of the functioning systems can be distinguished.

The main aim of the group control system (GCS) is the distribution of goals between mobile robots and planning the movement of nodes to their goals [8]. The scheme of the GCS is presented in Figure 1a. The mobile robotic control system is presented in Figure 1b. 

The task selection unit is responsible for assigning goals to each node. Nodes that receive the same goal are combined into one cluster. Regardless of the goal selection algorithm used, the goal selection unit relies on information received from other C_R_ nodes [9]. 

The formation unit and path planning unit determine the position of the nodes in space, relative to other nodes at certain points in time. The clustering unit is responsible for establishing communication within the cluster. 

The lower-level unit control performs the interaction of the robot control system with the execution mechanisms. 

Thus, the main objectives of this paper are as follows:Development of a wireless network model for a group of robots and simulation of group control.Development of methods for detecting abnormal behavior based on the analysis of cyber parameters and in-depth traffic analysis.

To achieve the above objectives, we executed the following steps:Analysis of structural and functional characteristics of robotic systems.Development of an experimental stand simulating the work of the robot group with the group control system.Development and implementation of attack scenarios for the robot group.Analysis of node characteristics that are changed under the influence of the attack.Development of a set of metrics that identify anomalous behavior.Software development for the implementation of the methodology.Analysis of the results of an experimental study, and determination of sets of metrics that are affected by a particular attack.

The article is organized as follows: Section 2 presents а review of papers on this topic; Section 3 describes the network model that was developed to collect statistical information; Section 4 contains a methodology for conducting attacks against the network in order to identify the parameters affected by the attack; Section 5 provides a method for identifying anomalies based on the collected statistical information and confirming the effectiveness of the method.

## 2. Related Work

To date, problems related to the safety of a group of mobile robots were actively studied. There were several studies related to the development of intrusion detection systems for mobile robots. The software has several requirements. 

The following metrics were used as the UAV attack behavior indicators in Reference [10]:The first indicator is that a UAV readies its weapon when outside the battlespace.The second indicator is that a trusted node’s embedded sensor reading differs from the monitor’s embedded sensor reading.The third indicator is that a monitor UAV provides bad-mouthing attacks, i.e., providing bad recommendations regarding a well-behaving trusted UAV.The fourth indicator is that a UAV deploys landing gear when outside its domestic air base.The fifth indicator is that a node sends bytes to unauthorized parties.The sixth indicator is that a UAV uses countermeasures without identifying a threat.The seventh indicator is that a loitering UAV uses more than the minimum thrust required to maintain altitude.

These indicators basically allow determining the mobile robot behavior from the position of the physical device. In this approach, there are advantages and disadvantages. The main drawback is that, if it is necessary to use this attack detection system for other types of mobile robots, then a set of indicators should be changed. 

The lack of connectivity parameter, which helps to detect a denial of service attack, was considered in Reference [11]. To identify attacks, the following metrics were used [12]: node reputation, behavior score, and distance estimation. Such metrics allow identifying attacks aimed at changing the behavior of nodes.

An intrusion detection system based on the C 5.0 decision tree for a group of robotic vehicles was considered in Reference [13]. The advantage of the presented approach was that the authors considered cyber-attacks. To detect these attacks, the authors used cyber input functions. They used four parameters to analyze the physical properties of the robot, which the authors called the physical characteristics of the input signal. Thus, they often investigated the behavioral characteristics of robots or changes in the physical environment.

In Reference [14] proposes the research of a technique for assessing the effectiveness of attacks on ad hoc networks. However, in this research, the authors mainly considered the attacks associated with the ad hoc on-demand distance vector (AODV) routing protocol. Although this protocol is often used to build ad hoc networks, there are other routing protocols, and their study was quite specialized. 

In Reference [15], the problem of detecting denial of service attacks using cyber parameter analysis was considered. The authors implemented a stand that includes a mobile guided robot. In this case, a denial of service (DoS) attack of the flood type was realized, affecting the movement of the robot. Two attack scenarios were developed. In the first case, the robot moved at the same speed, and, in the second case, it could accelerate and slow down. The impact of the DoS attack on the movements of the mobile robot was revealed as a result. Thus, the authors used the Snort attack detection system and increased two parameters: robot halting and delay in responding to navigation commands.

An algorithm for the sustainable formation of robots based on a multi-agent approach was presented in Reference [16]. To achieve resilient control, when an attack was detected, the state of the model was used to determine the control action for the agent, and the corrupted information coming through the network was ignored. In an experimental study, they demonstrated that robots managed to maintain formation even in the presence of DoS attacks by developing an algorithm. In this case, the robots managed to maintain the deployment regardless of whether the attack was implemented on one or several nodes or not. Unfortunately, the process via which this effect was achieved was not fully understood, and the description of the denial of service attack was not given.

The following subtypes of robotized systems were defined in Reference [17]: multi-agent robotized systems, mobile sensor networks, MANETs, robotized systems with swarm intelligence, and remotely controlled robots. Each system has its own features from the security point of view, which is why it is important to find a unique approach to and consider features of every system.

Zikratova et al. [18] considered the problem of building protection mechanisms for multi-agent robotic system protection from malicious robots. This paper was devoted to mechanisms of “soft” protection or second defense line. The authors constructed the security system with methods which are used in multi-agent computers (MAC). The authors claimed the applicability of Xudong’s method [19] and buddy security model (BSM) [20,21] for MAC protection, which agreed well with decentralized system architecture. In addition to that, they applied social control mechanisms, such as trust and reputation control, for user protection.

Feng et al. [22] described their approach based on reputation and trust. They proposed two algorithms to calculate the trust level between nodes. The nodes formed the estimation vector and generated a member’s estimation array. These estimations were formed according to the validity of the information and the distance to the target. As a result, the reputation function was described by Weibull–Gnedenko distribution. The drawback of this approach was caused by the fact that trust was calculated at a particular moment of time and for a particular problem. In addition to that, only one parameter was used, namely, the distance to the target. If an intruder carried out a number of various active attacks, which were not connected with data precision and accuracy, this system would not be able to detect that.

Vikshin et al. [23] proposed an approach to algorithm stability estimation based on trust and reputation. The authors used the following parameters for the estimation: interaction radius, number of agents, and percentage of adversaries in the total number of agents. The location of robots was chosen randomly in each experiment. The results reflected the percentage of correctly recognized legitimate agents and malicious agents in the entire network and within the interaction radius. As a result of the experiments, the authors confirmed that the trust model can successfully oppose intruders in any quantity. The main drawbacks of this approach to the network security estimation were the insufficient number of estimation parameters. From the presented parameters, one could only recognize the type of the intruder. The intruder model was described for a certain type of security violation only.

An approach to the detection of the malicious node according to the node’s behavior was developed in Reference [24]. In this approach, a group of robots followed a set of rules when fulfilling their mission. Then, the robots observed the behavior of others and exchanged relevant information. The behavior of the robots was described by a formal model proposed by the authors. The approach was based on the principle of using a protocol that allowed the robots to detect abnormal behavior of the others under certain conditions and to modify their own behavior to protect other robots from malicious activities. The protocol had two major components. The first one was a monitor of neighbor behavior, comparing it with the agent’s own behavior. The second one was the algorithm that allowed combining “opinions” of different monitors by communication. The authors carried out the experiments, demonstrating that at least six robots had to monitor the situation to detect intruders properly. The drawback of this method could be summarized as follows: when the system was detecting intruder, it relied only on the behavior of a node, which was characterized by its motions along a trajectory. That is, other types of attacks were not detected. To properly track an intruder, the system should have special monitor nodes, which track the network behavior, and their quantity must be at least six. This fact brought extra limitations to the network. Also, the authors did not analyze the effects of variability in the number of intruders and the top quantity of deflected intruders.

A system for detecting anomalous behavior of the internet of robot network was discussed in Reference [25]. This system was that it had two subsystems. The first system was a group of robots that collected data using a sensor system and transmitted it to a central server, which was connected to an external network. The second one was a mobile network, where the following modules were available: data acquisition module, anomaly classification module, and order module. The disadvantage of this system was that it was completely centralized; the robots did not communicate with each other and acted only through an intermediary. Anomaly detection was performed using a classifier that was trained using a pre-formed training sample. 

Thus, the following conclusions can be made:The topics of robotized systems security are the focus of research worldwide. Authors offered various approaches to robot security.There are no complex approaches to security of mobile robot networks on the network and physical layer, as well as on the control level [26].

The works discussed above were based on the need to compare the behavior of the robot with a reference one or the presence of an external observer, or to consider detection systems based on rules or signatures. In this case, the parameters analyzed to identify anomalies were very specific and may not be suitable for all systems. 

Our contributions to the state of the art are summarized as follows:Analysis of various architectural solutions, protocols, and standards for building a communication infrastructure for a group of robots.Analysis of the main approaches to the creation of group control systems for robots, and developing a group control algorithm for the experimental stand.Design and implementation of an experimental stand simulating the work of a group of robots.A methodology for in-depth traffic analysis based on mathematical statistics for a group of robots.We prove the effectiveness of using metrics, based on which the further development of an anomaly detection system in a robotic system is planned. As a result, the method allows determining the presence of an attack rather quickly and with a minimum loss of resources. This method takes into account the features of robotic systems and will be effective precisely for such systems.A software tool that implements the method of deep traffic analysis is developed and tested at the experimental stand.We conduct an experimental study proving the effectiveness of detecting anomalous behavior of a node based on a deep analysis of traffic, and we also confirm the low cost of resources for implementing calculations by network nodes.

## 3. Network Model for a Group of Mobile Robots

Figure 2 shows the network diagram implemented for conducting experimental research and collecting statistical information. It was assumed that the network node was a mobile robot. The mobile robot was a land mobile, which performed the task of movement toward the goal of a certain trajectory. The central server, a node with IP address (Internet Protocol Address) 192.168.1.2, performed the following tasks: providing goals for the mobile robots, and accumulating information. The monitoring mode for the Wi-Fi module of the central server was configured to implement these functions. The wireless mesh network was configured between nodes. The Wi-Fi modules of mobile nodes were configured in ad hoc mode to realize the mesh network topology.

Network nodes exchanged packets over wireless communication channels, and the 802.11 n standard was used [27]. This standard allowed setting up an ad hoc network between nodes. Routing protocols, such as AODV and optimized link state routing (OLSR), allowed extending the functionality of the ad hoc network and led to the mesh network creation. Network nodes could communicate with each other directly and, if necessary, via a neighbor through routing protocols. The connections between nodes in Figure 2 only partially show the complete communication scheme. In fact, nodes could communicate with each other through the routing protocol. In this network model, the OLSR routing protocol was used. The UDP protocol was used to transfer data at the transport layer [28]. The TCP protocol was used to transfer control commands and critical information.

Nodes with addresses 10.0.0.1–10.0.0.10 formed a segment of the mesh network, and node 192.168.1.2 was a central server which received all traffic that transmitted over the mesh network. In addition, node 192.168.1.2 transmitted control commands to the network nodes. 

Also, three types of nodes were assumed according to their behavior in the network:Trusted node—network node that is not conducting an attack and is not under attack. It performs its functions properly.Malicious, attacker, intruder node—network node that is conducting the active attack. It could be the internal or external network node.Victim node—network node that is under attack from the malicious node.

Nodes received targets from a central server and then distributed them among themselves. The group control subsystem was implemented to solve the problem of goal distribution. The goal distribution algorithm was chosen in Reference [29]. The main steps of the algorithm are presented below.

Step 1. Detection optimal plan for the goal distribution via a pairwise exchange between robots. At this step, the robots “changed” in pairs of goals to find out whether the replacement of the goals would lead to an increase in the functionality.

Step 2. Improving the plan via the pair-wise exchange of the goals. After completion of all the replacements, steps 1 and 2 were performed again successively, until all values at the next iteration of the execution of step 1 were obtained.

Then, step 3 was performed.

Step 3. The meaning of this step was to try each robot to improve their choice of goals by organizing a sequence of substitutions of goals. To do this, each robot determined whether replacing its goal with another goal improved the quality of the chosen target, performed until the most profitable target was chosen.

Step 4. Determination of the sequence of possible goal replacements in the robot group from all values. The maximum absolute value was selected and transmitted to all other the group of robots. Then, the feasibility of replacing goals with robots was determined. To do this, the robots calculated the improvement of the functionality in this step. 

Such a replacement was appropriate in the case when the evaluation of the effectiveness of the candidate goal was greater than the sum of the efficiency scores of the previously chosen goal and candidate goal for the other robot chosen by this robot.

## 4. Method for Identifying Anomalies

The methodology involved the steps described below.

1. Calculation of the total amount of network packets that went through the node:(1)$$\begin{array}{l} L_{Total}=\sum_{N}s_{data}(\Delta t)+\sum_{N}s_{routing}(\Delta t)+\sum_{N}r_{data}(\Delta t)+\sum_{N}r_{routing}(\Delta t)+\\ +\sum_{N}d_{data}(\Delta t)+\sum_{N}d_{routing}(\Delta t)+\sum_{N}f_{data}(\Delta t)+\sum_{N}f_{routing}(\Delta t) \end{array},$$
where *s_data_* is the total number of segments sent at the transport level, *s_routing_* is the total number of packets sent by routing protocol, r*_data_* is the total number of segments received at the transport level, r*_routing_* is the total number of received routing packets, d*_data_* is the total number of discarded segments transmitted at the transport layer, d*_routin_* is the total number of discarded routing packets, f*_data_* is the total number of forwarding segments at the transport level, f*_routing_* is the total number of forwarding routing packets, and *N* is the total number of nodes in the network.

It was necessary to evaluate the growth of this metric. If there is a rapid growth of the number of packets in the network, it is likely to be attacked.

2. The second step was to estimate the degree of variance of the network load parameter in past intervals relative to the average value. For this, it was necessary to calculate the general variance for the node network load value at the current interval for each node of the network:(2)$$D_{Lg}=\left(\sum_{i=1}^{N}{\left(L_{i}-{\bar{L}}_{g}\right)}^{2}\right)/N,$$
where *D_Lg_* is the general variance of the network load parameter, $\bar{L_{g}}$ is the general average for the network load factor, L_i_ is the load level of the current node, and *N* is the total number of nodes in the network. 

If the growth of a network metric values is observed, it can be concluded that an anomaly is present on the network.

3. Calculation of the ratio of sent to received packets:(3)$$Ratio_{s,r}=s_{i,n,data}/r_{i,n,data},$$
where *Ratio_s,r_* is the ratio between received and sent packets.

In normal network operation, the number of received data packets should approximately coincide with the number of sent packets, and there may be slight deviations. Therefore, if the ratio of sent and received packets is measured, then anomalous behavior of the node can be detected. If a node becomes a victim of a denial of service attack, then it receives more packets than it sends, or than other nodes receive.

4. Calculation of the deviation from the overall average for each network node:(4)$$Dev_{N,i,}{}_{s}(\Delta t)=(s_{n,i}(\Delta t)-\bar{s_{i}}(\Delta t))/N,$$
(5)$$Dev_{N,i,}{}_{r}(\Delta t)=(r_{n,i}(\Delta t)-\bar{r_{i}}(\Delta t))/N,$$
where $Dev_{N,i,}{}_{s}$, $Dev_{N,i,}{}_{r}$ are the deviations from the overall average parameter for the number of sent and received packets calculated for each node for the current time interval, and $\bar{s_{i}}(\Delta t)$, $\bar{r_{i}}(\Delta t)$ are the mean values for the number of packets received and sent for the current time interval for all nodes.

Analysis of this parameter does not give an unambiguous answer for which node is implementing the attack. However, it can serve as a good tool for determining the degree of anomalous activity of malicious nodes.

5. Calculation of the ratio of dropped packets in the current time interval from normal network operation:(6)$$Ratio_{i,d}(\Delta t)=d_{i}(\Delta t)/d_{norm_{i}}(\Delta t).$$

6. Evaluation of theratio for the sent, received, and dropped packets:(7)$$Ratio_{sent}=\frac{\left(\sum s_{total}(\Delta t)\right)}{L_{Total}}*100\%,$$
(8)$$Ratio_{received}=\frac{\left(\sum r_{total}(\Delta t)\right)}{L_{Total}}*100\%,$$
(9)$$Ratio_{dropped}=\frac{\left(\sum d_{total}(\Delta t)\right)}{L_{Total}}*100\%,$$
where *Ratio_sent_* is the percentage of sent packets in the total traffic, *Ratio_received_* is the percentage of received packets in the total traffic, *Ratio_dropped_* is the percentage of dropped packets in the total traffic, *s_total_* is the total number of sent packets for the current time interval, *r_total_* is the total number of received packets for the current time interval, and *d_total_* is the total number of dropped packets for the current time interval.

7. The monitoring of energy consumption was carried out using a KEWEISI KWS-V21 tester on a housing chip with an operating voltage of 3–20 V and a working current of 0–3.3 A. This tester was equipped with a bright screen and had the ability to display the following characteristics: voltage (U), current energy consumption (W), consumed battery capacity (mAh), and time (by a timer).

8. Oversupply of dropped packets (ODP). Under normal network operation, hosts can drop packets as a result of collisions and packet queue overflow. Thus, in normal network operation, when an intruder does not conduct an attack, the allowed number of dropped packets is 1%.

For initial verification of this metric, it was necessary to determine whether the number of dropped packets increased in comparison with the normal network operation for the current time interval:(10)$$ODP_{i}=d_{i}/d_{norm_{i}}.$$

If dropped packets exceeding the level were seen more than three times, the presence of abnormal activity in the network was considered. Dropped packets can occur not only during a black hole attack but also during a denial of service attack, since the nodes do not have time to process the packets and discard them from the queue.

9. Oversupply of forwarded packets (OFP). An intruder, performing some attacks, creates fake nodes or simulates the presence of a large number of malicious nodes in the network. For example, the main purpose of the Sibyl attack is to redirect the attacker’s influence on himself. It creates conditions such that packets should be routed through the fake nodes. Thus, there is a tendency to increase the number of routing packets. To determine the presence of anomalous activity during the attack, it is advisable to track the growth of forwarding packets. This can be done in two ways. Firstly, the level of forwarding of packets over the current time interval over the number of forwarding packets recorded during normal network operation is estimated:(11)$$OFP_{i}=(\sum_{N}f_{i,aodv};f_{i,aodv})/(\sum_{N}f_{i,}{}_{norm,aodv};f_{i,norm,aodv}),$$
where *OFP_i_* is the ratio between the number of forwarding packets at the current time for the node to the total number of forwarding packets per time interval for normal network operation.

Also, detection of abnormal activity can be measured by the percentage of the number of packets routed as a function of the total number of network packets:(12)$$Ratio_{f}=\frac{\left(\sum f_{aodv}+\sum f_{cbr}\right)}{L_{Total}}*100\%.$$

## 5. Experimental Results

To conduct an experimental study, a software tool was developed and implemented. Since most operating systems for mobile robots are Unix-like, we took the following steps:The tcpdump program was used as a network sniffer, since it has a powerful traffic interception tool, a lot of conveniently adjustable filters, and a console interface, which is convenient if you connect to a mobile device using remote access protocols; it is not exhausting many resources.The programming language was Python since it is supported by the operating systems of the Unix family by default, and it is interpreted, which eliminates hardware dependency as opposed to compiled ones.

### 5.1. Traffic Analysis Software

Functional requirements and steps were as follows:Call the network sniffer tcpdump from the main thread by a system call to intercept traffic on the node and then save the result to a file with the pcap permission.Analyze the received file, highlighting the information that will generate statistical data for further calculation of the main indicators of the network’s work.Carry out the calculation of network performance indicators; compare them with acceptable values obtained in the study of a normally functioning network.In case of deviation of the obtained values of indicators, notify the user about the abnormal behavior of the node, about the type of possible attack.Save to a text file all the collected statistical information, and calculate values of indicators for subsequent analysis of the incident by the user.

The software tool analyzed the parameters, calculating the metrics presented in the methodology. Firstly, metrics were analyzed during normal network operation. Through the normal operation of the network, it was understood that the nodes exchanged packets according to a predefined algorithm corresponding to the stages of the group management system, and this network was accessible. During the normal operation of the system, metrics were calculated.

The purpose of conducting attacks is to identify which metrics will change their values under the influence of the attack. During the experimental study, the intensity of the attack was changed. That is, the number of victims, the attack speed, etc. were changed depending on the type of attack. The main objective of the experimental study was confirming the effectiveness of the proposed metrics in identifying various types of attacks and determining which sets of metrics and which changes in which metrics are characteristic for which attacks. In addition, the goal was to determine how much the calculation data affects the performance of the device’s operating system.

### 5.2. Implementation and Analysis of the Impact of Attacks

#### 5.2.1. Denial of Service Attack 

The SYN flood (SYN - Synchronize sequence numbers) attack was chosen as an internal denial of service attack for the experimental study. This choice was due to the popularity of this attack in computer networks according to Kaspersky Lab [30]. The result of this attack was obviously a noticeable increase in resource consumption by the device that was attacked. During the attack, there was no availability of the communication channel.

Three attack scenarios were developed. The first scenario was that one intruder node actively attacked one trusted node such that there was one victim in the network. In the second scenario, an intruder attacked 25% of the trusted nodes. In the third scenario, 50% of the nodes were victims. Since, out of the 10 nodes, one was an internal attacker and one node was a central server, the principle of the attack was that the attacker sent SYN requests and overflowed the connection queue for the victim of the attack. Half-open connections appeared in the queue, waiting for confirmation from the client.

Figure 3 shows the change in the variance of the network load for nodes *D_Lg_* by Equation (2). It can be seen from the figure that the variance increased significantly upon increasing the number of victims. This means that the load of nodes was quite different from the average. This is because the load level of the malicious node was significantly greater than the node for the load of trusted hosts. The load level of the victim nodes also increased in comparison with the normal state. During normal network operation, when nodes sent packets according to a given algorithm and operated in the normal mode, the dispersion level could reach 7–10. Thus, it can be said that there was abnormal activity in the network.

Three types of situations were simulated. In the first case, the estimation of the energy consumption by network nodes in the absence of an attack was carried out. In Figure 4, this situation is represented by a blue chart marked with rhombuses. In the second case, the network was attacked, while the traffic of the malicious node was I_t_ ≤ I_m_ ≤ 2I_t_, where I_m_ is the malicious node traffic, and I_t_ is the traffic of the authentic node. 

The third graph represents a situation where an attack was carried out intensively and I_m_ > 2I_t_ [31]. Figure 4 shows that, during the low-intensity attack, the energy level of the nodes remained almost the same as for the case when the attack was not carried out. That is, in this case, the attack could be considered ineffective. The graph reflects the change in the energy level during intense attacks where a sharp drop in energy level was seen, which confirmed the effectiveness of the attack. At the same time, the load of the attacker’s node was more than twice the workload of trusted nodes. The effectiveness of this attack could also be assessed by the presence of discarded packets. Malicious nodes create such a situation in the network when trusted hosts cannot take a large number of requests and discard packets from the queue. Together with packages from malicious nodes, the victims also discard useful packages. Figure 5 shows that a more intense attack with more intruders on the network resulted in an increase in the percentage of dropped packets. Thus, the sender does not re-send the necessary information, and the recipient does not know that the information was transferred to him [32]. 

The presence and effectiveness of a denial of service attack are fairly accurate and can be effectively determined using the ratio of sent to receive packets. Figure 6 shows that the nodes under attack took more packets than they sent, with a difference of more than 2–2.5 times. In this case, the attack was considered effective [33].

As a result, the attack affected the energy consumption of the node. The result of this attack was obviously a noticeable increase in the resources consumed by the device that was attacked. During the attack, there was no availability of the communication channel. Figure 7 shows the result of attacks of varying degrees of intensity on the experimental stand. As can be seen from Figure 7, as the intensity of the DoS attack increased, so did the number of resources consumed by the node. The figure shows the results for a node that was the victim of an attack.

One of the main indicators of the effectiveness of an attack is a change in in the energy consumption of the nodes. A sharp change in this indicator shows that the node is spending resources on maintaining a large number of network connections. Figure 7 shows the situation where energy consumption was 1.5 times greater than in a situation where the attack was not conducted. 

Figure 8 shows the change in the central processing unit (CPU) indicator for the normal state of the node when there was no attack on the node, and for the victim’s node, on which a low-intensity attack and a medium-intensity attack were carried out. Figure 8 shows that, in the first minutes of the attack, there was a sharp increase in CPU utilization, followed by no sharp increases; however, CPU utilization remained at a high level. Analysis of this parameter can be used to implement the host-based intrusion detection system.

#### 5.2.2. Black-Hole Attack

The Black-Hole attack involves the intruder located between two trusted nodes and, instead of forwarding packets, they are dropped. As a rule, the nodes located at a considerable distance from the base station or the group leader can fall victim to such an attack.

When the experimental stand was developed, one device was the server and other devices were the clients. For each of them, corresponding sockets (server, client) were written in the Python language. The client sent the string packet to the server, and the server sent it back to the client. The attacker’s goal was to intercept the data sent by the server. An ARP-spoofing attack was implemented first to intercept traffic.

If the attacker needs to perform the Black-Hole attack, then the Ip table rules for the malicious node should be configured so that the attacker drops packets passing through it (Ip tables is a user-space utility program that allows a system administrator to configure the tables provided by the Linux kernel firewall).

During the Black-Hole attack, the variance level was *D_Lg_* = 829.3, which greatly exceeded the variance under normal network conditions. This indicated abnormal activity in the network. An efficient parameter in the detection of this attack is to estimate the growth of dropped packets (see Figure 9). It can be seen from the figure that, as the number of malicious nodes increased, the number of dropped packets increased, in comparison with the normal operation of the network. 

This suggests that a greater number of nodes were attacked and isolated from the central server.

In addition, there was a decrease in the number of received packets, due to the fact that the attacker dropped packets sent through it by the node and the response did not come to the node. An evaluation of the ratio between packages of different types was quite indicative. During normal operation of the network, the ratio of sent and received packets was observed in equal parts, with a small amount presenting dropped packets (see Figure 10). Also, from the figure, it can be seen that, as the number of malicious nodes increased, the traffic pattern changed. The ratio of the number of sent and received packets was no longer equal. This observation can be useful in developing an attack detection system. As for the level of energy consumption, it did not change significantly. It can be said that even lower consumption was observed, since the victim’s node did not spend energy receiving the package.

#### 5.2.3. Deauthentication Attack Creating Fake Clients and Connections to the Access Point

A deauthentication attack is considered as a variant of an external denial of service attack. The implementation of the deauthentication attack is possible using aireplay-ng. Aireplay-ng is a tool that includes aireplay-ng deauthentication, such as airgeddon. To conduct this attack, one firstly needs to determine on which channel the access point operates. 

The result of this attack was disconnecting all wireless clients from the network that were connected to it, regardless of how the attack was implemented. It follows that the attack was successful since its goal was achieved.

During 10 min of the attack on the mobile robot, 523,000 authentication requests were sent. After 17–20 s from the beginning of the attack, the client of the network was disconnected. After deauthentication, the client constantly tried reconnecting; however, since the access point could not handle a large number of authentication requests, the establishment of the connection was almost unsuccessful. If the client was able to connect, then, after 2–3 s, deauthentication occurred, and it tried connecting again. Also, this attack was performed on a mobile device (cell phone), which acted as an access point. After 40–50 s of the attack, the device disabled the access point. It follows that the response to this attack by different devices may differ. Based on the data obtained during the experiments, shown in Figure 11 and Figure 12, it can be seen that the attack was resource-intensive.

During the attack, 116 mAh of battery capacity was expended; thus, for greater autonomy and maintaining mobility, a portable battery with a capacity of 10,000 mAh could be used. In addition, the figure shows that the power consumption of the victim of the attack also increased compared to the normal state. This is because the victim was forced to reconnect to the access point and spend additional energy on this process.

Figure 12 shows that this attack expended additional computational power not only for the attacker but also for the victim of the attack. 

Table 3 shows the attacks and their influence on the characteristics of the mobile device.

From Table 3, it can be seen that the characteristics most affected by the attacks are the CPU load, power consumption, and channel availability. The influence on the characteristics of the processor load and the energy consumption is, for the most part, due to attempts to reconnect the client to the attacked network during deauthentication attacks. When SYN-flood is realized, due to the large amount of incoming traffic, CPU load is increased. The impact on the availability of the channel, with external attacks, is the impossibility of connecting to the access point of the wireless network for several reasons, depending on the reaction of the access point to the attack.

### 5.3. Analysis of the Results of an Experimental Study

Therefore, to summarize the experimental study, we can consider Table 4. Table 4 presents the control numbers of the experiments. After conducting various types of attacks with varying degrees of intensity, several findings were determined. The data presented in the table were obtained from the node, which was the attack. That is, the victim node performed the calculations itself and relied on the data collected by it. Table 4 shows that, in denial of service attacks, a significant increase in the metric dispersion of network congestion is observed. When Black-Hole attacks occur, there is a decrease in the obtained metric values compared to normal behavior. The decrease in the metric values is due to the fact that, during the Black-Hole attack, the traffic intensity decreases. An attacker drops packets and, therefore, proxies do not receive them. Therefore, the nodes do not spend additional resources for receiving packets, and the overall network load is reduced. Denial of service shows a sharp increase in traffic, and the Black-Hole decreases in traffic levels. The *Ratio_s,r_* metric should tend to one; with a denial of service attack, an increase in the metric is observed, and, with the Black-Hole attack, it goes to zero. For ${\mathrm{Dev}}_{N,i,}{}_{s}$ in the examples of the presented attacks, we can say that, if an attack causes traffic growth, then the metrics grow; if there are different attack intensities, then the metrics grow to one degree or another. If the attack reduces traffic, then the metrics tend to zero or less, compared with the nominal behavior. Thus, we can say that the selected metrics allow us to fix the deviation from the static behavior of the system, while it can change both in the direction of traffic growth and decrease it. Also, we added ICMP-flood (ICMP - Internet Control Message Protocol) attack.

According to the test results, it was determined that the developed software allows you to analyze the behavior of the device on the network, identify anomalies, and determine what type of attack is carried out on the network node. The maximum optimization of the application was achieved, for which the consumption of hardware resources was minimal.

A distinctive feature of the application is the ability to test the site autonomously, not based on the analysis of neighboring nodes.

## 6. Conclusions

In this article, various types of attacks on groups of mobile robots were considered and realized. Many attacks include several stages. For the majority of attack prevention methods were presented. The most frequent attacks are related to the possibility of unauthorized access to the communication channel, as well as vulnerabilities of data transfer protocols. At the same time, the denial of service attacks can lead to full network capacity, and then to the inability of the system to perform its functions. The second most dangerous class of attacks is directed at interception and sweeping information, such as a man in the middle. These types of attacks were implemented at the experimental stand, which included single-board computers and used a wireless communication channel for transmitting data. 

The developed set of metrics provides an in-depth traffic analysis in the group of moving objects, as well as the parameter of energy consumption (Table 5). Table 5 shows the set of attacks defined for each metric. Table 5 shows what they were able to identify. In future work, it is planned to use these metrics to calculate the trust value. In previous studies by the authors, three metrics were used to calculate the node trust indicator: network load, residual energy, and dropped packets. Using these metrics, the network load in the form of a set of metrics presented in this paper, with more detailed analysis of traffic, will increase the range of attacks that the system can counteract and also improve the accuracy of detecting untrusted nodes. One of the main differences between this study and the existing ones is that, unlike the works discussed in Section 2, this technique allows the node to analyze its behavior. At the same time, the developed software demonstrates low CPU consumption. CPU utilization with the running program was only 4%. In addition, the analyzed metrics proposed in this study are universal; they do not depend on specific protocols or the operating conditions of the system. These metrics could be used in a further study by the authors. It is planned to use the mathematical apparatus of probability theory to determine the degree of deviation of the metrics obtained at a given time from the confidence interval.

It is also planned to carry out normalization of the obtained values and analyze changes in the results of the wall function. At the moment, it can be said that the contribution of this study is that new dependencies were identified in the analysis of traffic changes, which will allow revealing anomalies and determining the type of attack. These dependencies are more characteristic for robotic systems since they are aimed at identifying deviations in the system’s operation, from accepted static and weakly modifiable changes in the system’s cycles.

## Figures and Tables

**Figure 1 sensors-19-04007-f001:**
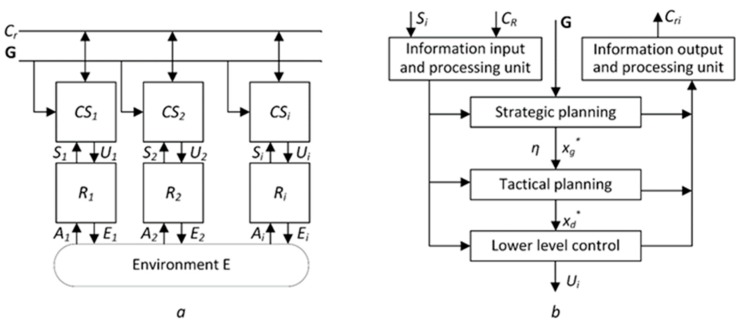
Group control system diagram: (**a**) The scheme of the GCS. G denotes the goals that are assigned to the group, Cr is the information received from other nodes, CS_i_ is the control system of robot R_i_, E is the environment in which the goals are performed; (**b**) The mobile robotic control system. C_R_ is the set of connections of the given robot with other robots of the group, x_d_ is the vector of the coordinates of the next point of the robots trajectory, and x_g_ denotes the coordinates of the chosen goal.

**Figure 2 sensors-19-04007-f002:**
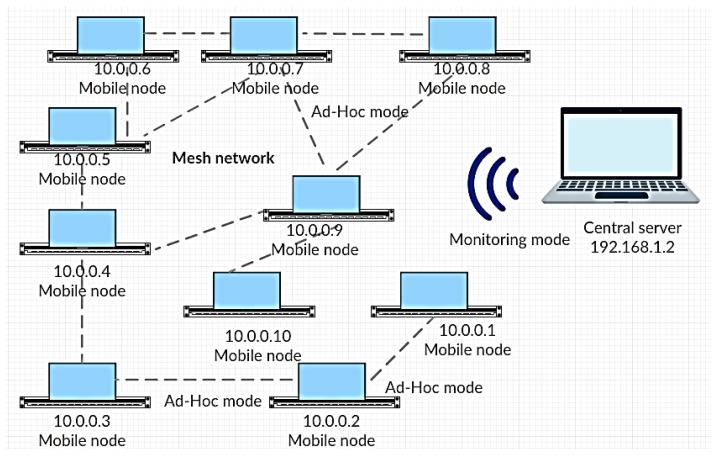
Model of wireless network for a group of mobile robots.

**Figure 3 sensors-19-04007-f003:**
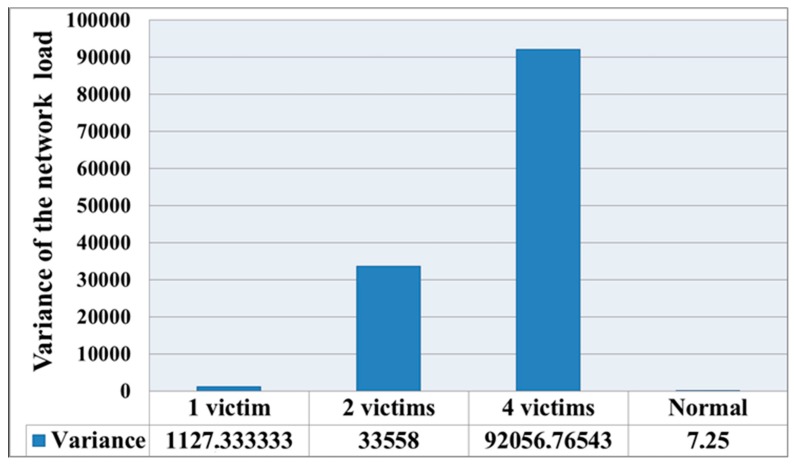
The variance of the load parameter *D_Lg_* for nodes under denial of service (DoS) attack with various numbers of victims.

**Figure 4 sensors-19-04007-f004:**
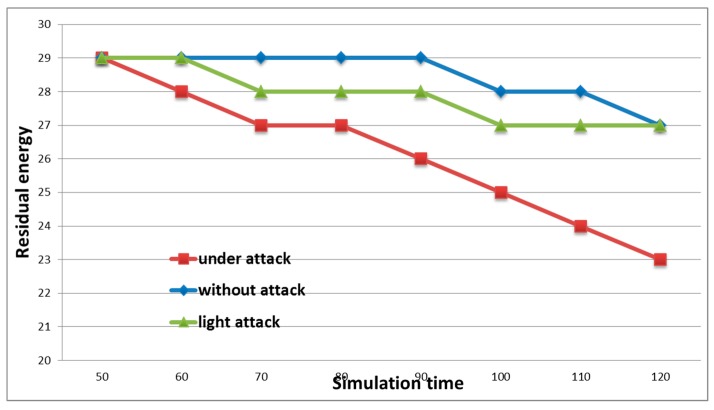
Changes in the residual energy level depending on the traffic load of network nodes.

**Figure 5 sensors-19-04007-f005:**
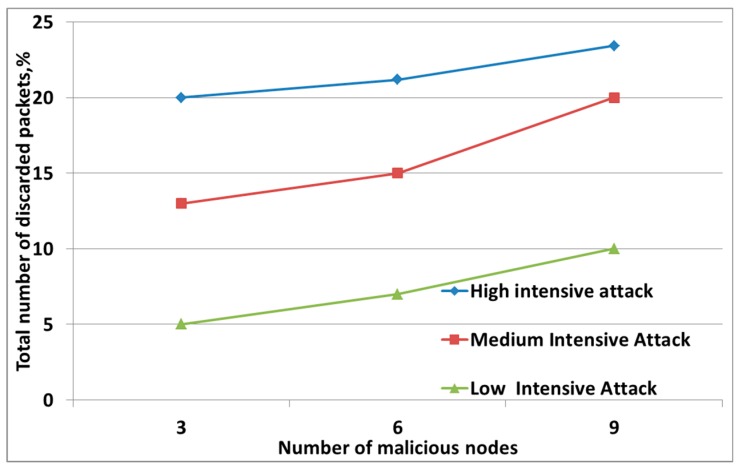
Changes in the number of dropped packets depending on the number of malicious nodes for attacks with varying degrees of intensity.

**Figure 6 sensors-19-04007-f006:**
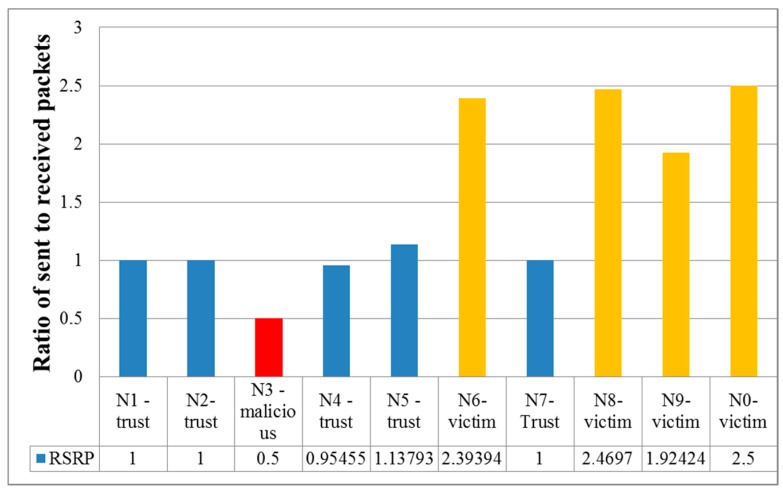
Ratio of sent to received packets *Ratio_s,r_* for trust, victim, and malicious nodes during DoS attack.

**Figure 7 sensors-19-04007-f007:**
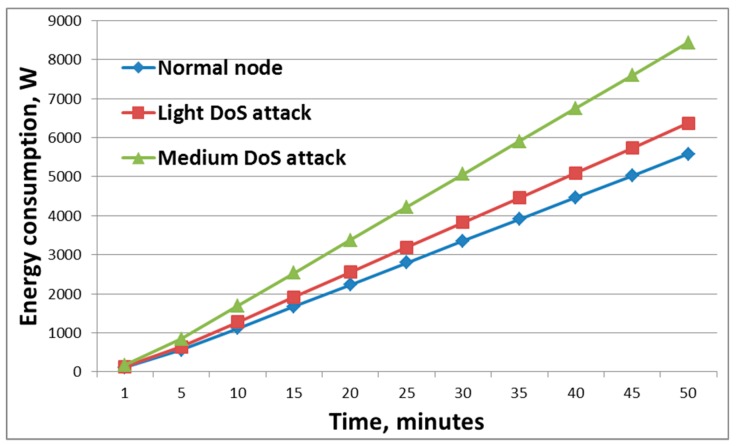
Energy consumption during denial of service attacks.

**Figure 8 sensors-19-04007-f008:**
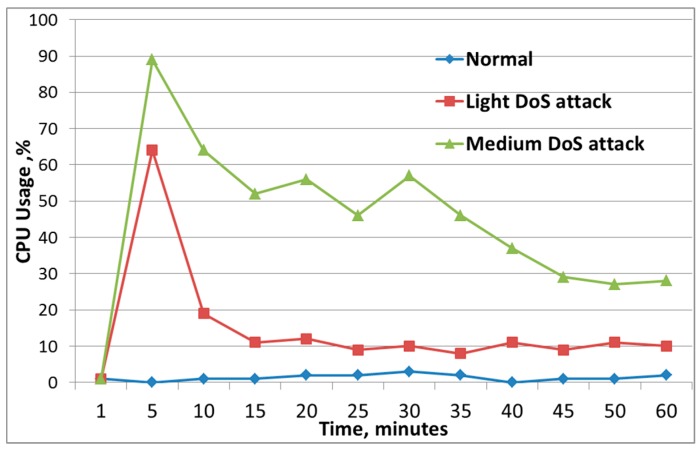
Changes in in central processing unit (CPU) load during denial of service attack.

**Figure 9 sensors-19-04007-f009:**
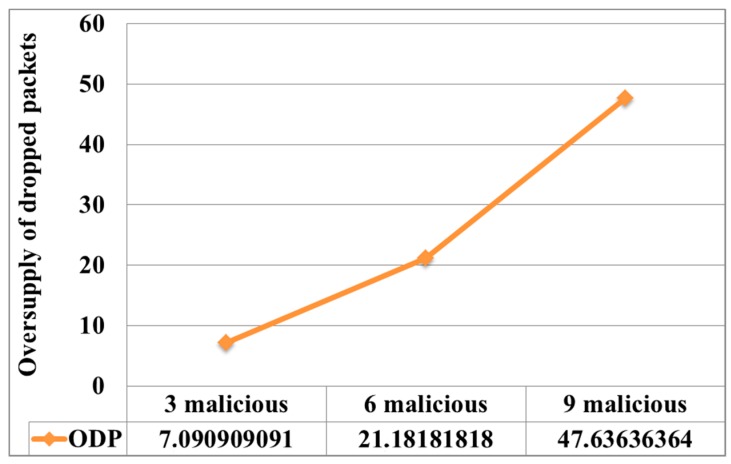
Change in the number of dropped packets for the network oversupply of dropped packets (*ODP_i_*) under the Black-Hole attack compared to normal network operation.

**Figure 10 sensors-19-04007-f010:**
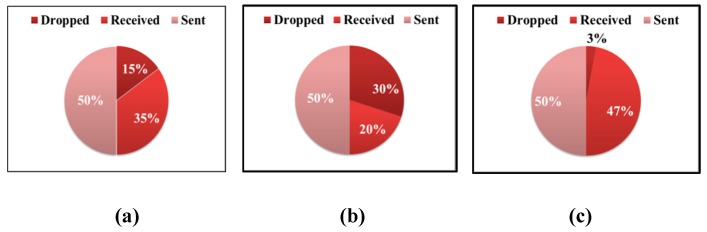
Changing the pattern of traffic for the network under the Black-Hole attack with a change in the number of malicious nodes: (**a**) for three malicious hosts; (**b**) for nine malicious hosts; (**c**) when malicious nodes were missing.

**Figure 11 sensors-19-04007-f011:**
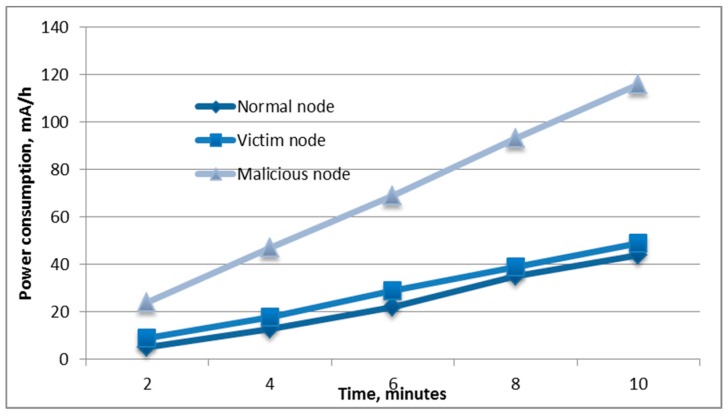
The power consumption of the nodes under the deauthentication attack.

**Figure 12 sensors-19-04007-f012:**
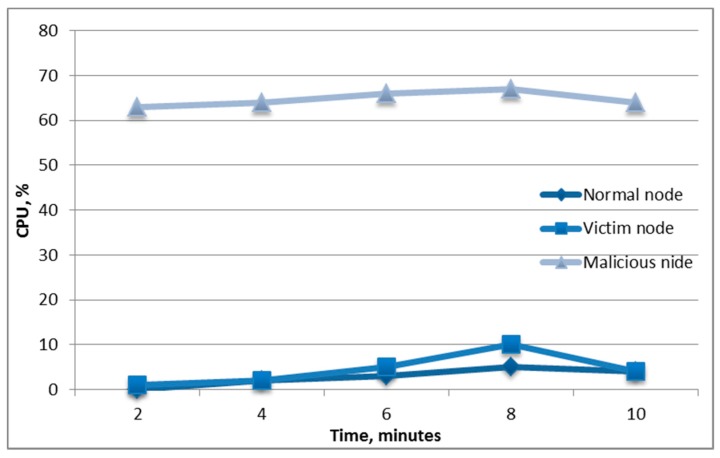
The CPU consumption of the nodes under the deauthentication attack.

**Table 1 sensors-19-04007-t001:** The types of Mobile Robots Control Method and ways of organizing and building the networks.

No.	Type of Mobile Robots Control Method	Features of Structural and Functional Characteristics
1	Remote control	The radio channel is established directly between the operator and the mobile robot. The mobile robot executes those commands that it receives directly from the operator.
2	Independent control	The mobile robot is pre-programmed and performs a specific sequence of actions. In addition, intelligent methods which allow the mobile robot to make decisions in unknown conditions could be used.
3	Overall control	It is implemented using a star topology, where a radio channel is organized to transmit a signal between a server and a mobile robot, usually using a wireless router or radio station.
4	Group control	It is organized using the ad hoc and mesh network. The wireless radio channel is used for communication between mobile robots. If it uses a fully distributed network structure, mobile robots could send packets to each other in multidirectional mode. Routing protocols are used to deliver packets to network nodes that are far apart from each other.

**Table 2 sensors-19-04007-t002:** Types of protocols for mobile networks.

No.	Network Level	Communication Protocols
1	MAC (medium access control) level	IEEE 802.11 n,s, ZigBee, LODMAC (Location oriented directional MAC protocol for FANETs).
2	Network layer	TBRPF (topology dissemination base on reverse-path forwarding), DSR (dynamic source routing), GPSR (greedy perimeter stateless routing), OLSR (optimized link state routing protocol), DOLSR (directional optimized link state routing protocol), AODV (ad hoc on-demand distance vector), HWMP (hybrid wireless mesh protocol)
3	Transport level	TCP (Transmission Control Protocol), UDP(User Datagram Protocol). TCP does not work well on mobile ad hoc networks (MANETs) and is not suitable for use in flying ad hoc networks (FANETs). JAUS (joint architecture for unmanned systems) is an emerging standard for messaging between unmanned systems.

**Table 3 sensors-19-04007-t003:** The impact of attacks on the characteristics of the mobile device. DDoS—distributed denial of service; CPU—central processing unit; RAM—random-access memory.

	Attack	DDoS	SYN-Flood	Black-Hole
Parameter				
CPU load (%)		+	+	-
Download RAM (Mb)		-	-	-
CPU temperature (° C)		-	+	-
Power consumption		+	+	-
Number of packages		-	+	+
Channel availability		+	-	-
Dropped packet		-	+	+

+ indicates that influence takes place; - denotes that no influence takes place.

**Table 4 sensors-19-04007-t004:** Additional research findings on metric changes for attacks and normal operation.

Attack	Metrics				
	*L_Total_*	$D_{Lg}$	*Ratio_s,rr_*	$Dev_{N,i,}{}_{s}$	$Dev_{N,i,}{}_{r}$
SYN-flood	275	7632	2.5825	8.375	14.125
ICMP-flood	645	51.266	22.6763	10.666	18.666
Black- hole	133	3540	0	−5.75	−9.0
Normal	144	455	0.98	−5.75	5.75

**Table 5 sensors-19-04007-t005:** Metrics for detecting attacks and evaluating their effectiveness.

Metric	Attacks		
	DoS	SYN-Flood	Black Hole
*D_lg_* (general variance of the network load)	+	+	+
Residual energy (RE)	-	+	*
CPU load (%)	+	+	-
*L_Total_* (total number of network packets)	+	*	-
*OFP_i_* (oversupply of forwarded packets (OFP))	+	+	-
*ODP_i_* (oversupply of dropped packets)	-	*	+
$Dev_{N,i,}{}_{s}$, $Dev_{N,i,}{}_{r}$ (deviations from the overall average parameter for the number of packets sent and received)	+	+	+
*Ratio_s,r_* (ratio of sent to received packets)	+	+	+

+ means that there is a clear sign of an attack and significant changes occur under the influence of an attack; - means that there are no changes in victim parameters when the node is under the influence of an attack; * denotes the indirect sign of an attack that may undergo minor changes.

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
