# Peer review of "Method for Detecting Abnormal Activity in a Group of Mobile Robots"

_sensors, 2019, doi:10.3390/s19184007_

Round 1

Reviewer 1 Report

The paper seems as an attempt to summarise the previous set of experiments, described in other articles by the authors (what is suggested by appropriate references, e.g., to [32] given it the text and the provided results). From this point of view, in this paper:

the experiments gathered are obtained not only from the network described in figure 2 but also from other configurations, (e.g., in figures 7 and 8 only two nodes were considered, which is not explained in the text)  it is not clear what is new in this paper, compared to the previously made analysis by the authors. There is a comment about the differences to the work of other authors (line 645), but what are the new results (if they are) in this paper? Probably the part with table 4 but what about the other charts and tables? if the authors planned to check the defined indices for the attack scenarios discussed in their previous work, it would be good to: exclude the indirect results that were obtained from other systems (i.e., for example fig. 7 and 8) (only important information should be mentioned and appropriate reference should be given) and focus on the results of the indices, preferably for higher number of different attack cases (also analysed in previous papers). Experiments with the system described by fig.2 should then provide exhaustive results for tables 4 (there are two tables 4 in the paper, why the first does not provide all the discussed in section 4 parameters?. Present results given in the paper seem as chosen at random.

There are also many inaccuracies in the paper. e.g.,:

in figure 6, we can observe sent/received ratio and the measure is defined otherwise; it would be also useful to use symbols (or acronyms) of measures if they are analysed in given figure (e.g., whether figure 3 shows us one of defined before measures?). from figure 7 it is not visible mentioned afterwards 20 times increase of energy consumption. the results presented in table 3 differ slightly from the results presented in similar table in [32], what is the reason for that? Also, "=/-" entry is never used (defined below the table). in section 4 - if 'normal' parameters are taken from experimental results, it should be stated where they are given (table 4?) or how they can be obtained. There is a lack of equation (10). the acronyms used in the last table should be introduced in the section where their definitions are given (section 4). Sometimes acronyms and, in other places, symbols are used for the same measure, what is confusing.

It is revised version of the paper and still there are many typos, repetitions of text, and other simple mistakes (not to mention grammar errors that can be more difficult to detect by authors).

Author Response

The paper seems as an attempt to summarise the previous set of experiments, described in other articles by the authors (what is suggested by appropriate references, e.g., to [32] given it the text and the provided results). From this point of view, in this paper:

the experiments gathered are obtained not only from the network described in figure 2 but also from other configurations, (e.g., in figures 7 and 8 only two nodes were considered, which is not explained in the text)  it is not clear what is new in this paper, compared to the previously made analysis by the authors. There is a comment about the differences to the work of other authors (line 645), but what are the new results (if they are) in this paper? Probably the part with table 4 but what about the other charts and tables? if the authors planned to check the defined indices for the attack scenarios discussed in their previous work, it would be good to: exclude the indirect results that were obtained from other systems (i.e., for example fig. 7 and 8) (only important information should be mentioned and appropriate reference should be given) and focus on the results of the indices, preferably for higher number of different attack cases (also analysed in previous papers). Experiments with the system described by fig.2 should then provide exhaustive results for tables 4 (there are two tables 4 in the paper, why the first does not provide all the discussed in section 4 parameters?. Present results given in the paper seem as chosen at random.

There are also many inaccuracies in the paper. e.g.,:

in figure 6, we can observe sent/received ratio and the measure is defined otherwise; it would be also useful to use symbols (or acronyms) of measures if they are analysed in given figure (e.g., whether figure 3 shows us one of defined before measures?). from figure 7 it is not visible mentioned afterwards 20 times increase of energy consumption. the results presented in table 3 differ slightly from the results presented in similar table in [32], what is the reason for that? Also, "=/-" entry is never used (defined below the table). in section 4 - if 'normal' parameters are taken from experimental results, it should be stated where they are given (table 4?) or how they can be obtained. There is a lack of equation (10). the acronyms used in the last table should be introduced in the section where their definitions are given (section 4). Sometimes acronyms and, in other places, symbols are used for the same measure, what is confusing.

It is revised version of the paper and still there are many typos, repetitions of text, and other simple mistakes (not to mention grammar errors that can be more difficult to detect by authors).

Dear Reviewer,

Thank you very much for reviewing our paper, and for your comments and suggestions. We tried to improve the paper as much as possible within the allotted time of 10 days. In accordance with your comments and suggestions, we have prepared our responses and made the corrections needed.

Reviewer 1

the experiments gathered are obtained not only from the network described in figure 2 but also from other configurations, (e.g., in figures 7 and 8 only two nodes were considered, which is not explained in the text)  it is not clear what is new in this paper, compared to the previously made analysis by the authors.

Response to Reviewer 1

We replaced the previous figures. The figures depict only three nodes so that it can be clearly seen, a change in the parameter for the intruder, the victim, and the nominal node. If you show the change in the parameter of each network node, then the picture looks bulky and oversaturated.

A previous study of the authors was carried out both for the model in the simulator and for the experimental stand. The data obtained do not contradict each other and confirm the logic of similar changes for nodes.

Reviewer 1

it is not clear what is new in this paper, compared to the previously made analysis by the authors.

Response to Reviewer 1

The previous study relied on studying only cyber parameters. In particular, only the denial of service attacks was studied. In this work, the spectrum of attacks was expanded. In addition, the range of parameters that were analyzed was expanded.

We add sections state of the art and main objectives

(line 107)

Thus, the main objectives of this paper are:

Development of a wireless network model for a group of robots and simulation of group control. Development of methods for detecting abnormal behavior based on the analysis of cyber parameters and in-depth traffic analysis.

To achieve the above objectives, we have executed the following steps:

Analysis of structural and functional characteristics of robotic systems. Development of an experimental stand simulating the work of the robots group with the group control system. Development and implementation of attack scenarios for the robots group. Analysis of node characteristics that are changed under the influence of the attack. Developing a set of metrics that identify anomalous behavior. Software development for the implementation of the methodology. Analysis of the results of an experimental study, determination of sets of metrics that are affected by a particular attack.

(line 259)

Our contributions to the state-of-the-art are summarized as follows:

Analysis of various architectural solutions, protocols, and standards for building a communication infrastructure for a group of robots. Analysis of the main approaches to the creation of group control systems for robots and developing a group control algorithm for the experimental stand. Design and implementation of an experimental stand simulating the work of a group of robots. A methodology for in-depth traffic analysis based on mathematical statistics for a group of robots. We have proved the effectiveness of using metrics, based on which the further development of an anomaly detection system in a robotic system is planned. As a result, the method allows determining the presence of an attack rather quickly and with a minimum loss of resources. This method takes into account the features of robotic systems and will be effective precisely for such systems. A software tool that implements the method of deep traffic analysis has been developed and tested at the experimental stand. We have conducted an experimental study proving the effectiveness of detecting anomalous behavior of a node based on a deep analysis of traffic, and also we have confirmed the low cost of resources for implementing calculations by network nodes.

Reviewer 1

There is a comment about the differences to the work of other authors (line 645), but what are the new results (if they are) in this paper? Probably the part with table 4 but what about the other charts and tables?

Response to Reviewer 1

A new result is a method for analyzing parameters. As well as implemented software, which proved the low cost of computing resources for the presented calculations? We have added in-depth traffic analysis.
The changes in processor congestion and energy consumption cannot always speak of a network attack. We've added a detailed study of traffic, to say unequivocally whether the passes of network attacks and to assume its type.

We changed figures 7,8 according new results. And also a new description was added to the figures.

(line 492)

Figure 7 shows the result of attacks of varying degrees of intensity on the experimental stand. As can be seen from Figure 7, then greater the DoS attack intensity, the greater the number of resources a node consumes. The figure shows the results for a node that was a victim of an attack.

(line 503)

Figure 8 shows the change in the CPU indicator for the normal state of the node when there was no attack on the node, for the victim’s node, on which a low-intensity attack and a medium-intensity attack were carried out. Figure 8 shows that in the first minutes of the attack there is a sharp increase in CPU utilization, then there are no sharp increases, but CPU utilization remains at a high level.

We revised table 5

line 641

Table 5..Metrics for detecting attacks and evaluating their effectiveness.

Metric

Attacks

DoS

SYN-flood

Black hole

Dlg (General variance of the network load)

+

+

+

Residual energy (RE)

-

+

*

CPU load (%)

+

+

-

LTotal (Total amount of network packets)

+

*

-

OFPi (Oversupply of forwarded packets (OFP)

+

+

-

ODPi (Oversupply of dropped packets)

-

*

+

,  (Deviations from the overall average parameter for the number of packets sent and received)

+

+

+

Ratios,r (Ratio of sent to received packets)

+

+

+

+ means that it is a clear sign of an attack and changes significantly under the influence of an attack;
- means that there is no changes in victim parameters when node was under the influence of an attack;
* denote an indirect sign of an attack that may undergo minor changes.

Reviewer 1

if the authors planned to check the defined indices for the attack scenarios discussed in their previous work, it would be good to: exclude the indirect results that were obtained from other systems (i.e., for example fig. 7 and 8) (only important information should be mentioned and appropriate reference should be given) and focus on the results of the indices, preferably for higher number of different attack cases (also analysed in previous papers).

Response to Reviewer 1

We tried to see the figures and graphs and provided new information.

We added the following

(line 569)

During the attack, 116 mAh of battery capacity was expended, so for greater autonomy and maintaining mobility, a portable battery with a capacity of 10,000 mAh could bue used. In addition, the figure shows that the power consumption of the victim of the attack also increased compared to the normal state. This is because the victim is forced to reconnect to the access point and spend additional energy on this process.

(line 594)

So, to summarize the experimental study let’s consider Table 4. Table 4 presents the control numbers of the experiments. After conducting various types of attacks with varying degrees of intensity, the following was determined. The data presented in the table were obtained from the node, which was the attack. That is, the victim node performed the calculations itself and relied on the data that was collected by it. Table 4 shows that in denial of service attacks, a significant increase in the metric dispersion of network congestion is observed. WhenBlack Hole is occur, there is a decrease in the obtained metric values compared to normal behavior. The decrease in the metrics is due to the fact that during the Black hole attack the traffic intensity decreases. An attacker drops packets and therefore proxies do not receive them. Therefore, the nodes do not spend additional resources for receiving packets, and the overall network load is reduced.

Reviewer 1

Experiments with the system described by fig.2 should then provide exhaustive results for tables 4 (there are two tables 4 in the paper, why the first does not provide all the discussed in section 4 parameters?. Present results given in the paper seem as chosen at random.

Response to Reviewer 1

The data presented in the table were obtained from the node, which was the attack. That is, the attack target performed the calculations itself and relied on the data that was collected by it. The implementation of the software is described above, and it is indicated there that the software module collects data from the victim. In previous works, the behavior of the network as a whole was analyzed.

Reviewer 1

5 in figure 6, we can observe sent/received ratio and the measure is defined otherwise; it would be also useful to use symbols (or acronyms) of measures if they are analysed in given figure (e.g., whether figure 3 shows us one of defined before measures?). from figure 7 it is not visible mentioned afterwards 20 times increase of energy consumption.

Response to Reviewer 1

The authors agree with the remark and made appropriate adjustments to the article.

Reviewer 1

the results presented in table 3 differ slightly from the results presented in similar table in [32], what is the reason for that? Also, "=/-" entry is never used (defined below the table).

Response to Reviewer 1

Table 3 shows the results for the victim of the attack. That is, it was estimated how the attack affects the parameters of only the victim. In a previous study, the authors examined the impact of attacks, including on the attacker. Therefore, differences can be observed.

We added the following (lines 584-585)

“+”indicates that influence takes place;

“-” denotes that no influence take place.

Reviewer 1

in section 4 - if 'normal' parameters are taken from experimental results, it should be stated where they are given (table 4?) or how they can be obtained.

Response to Reviewer 1

Normal behavior was observed for several hours at the experimental stand. At the same time, the nodes of the experimental stand exchanged packets according to the indicated protocols and performed sequentially the stages of the task of group management (according to the script described in the article). Thus, the nodes emulated the nominal operation of the system.

We added

(line 596)

The data presented in the table were obtained from the node, which was the attack. That is, the victim node performed the calculations itself and relied on the data that was collected by it.

Reviewer 1

There is a lack of equation (10). the acronyms used in the last table should be introduced in the section where their definitions are given (section 4). Sometimes acronyms and, in other places, symbols are used for the same measure, what is confusing.

Response to Reviewer 1

The authors accept the comment and revised these shortcomings in the article.

We added the equation.

Reviewer 1

9 It is revised version of the paper and still there are many typos, repetitions of text, and other simple mistakes (not to mention grammar errors that can be more difficult to detect by authors).

Response to Reviewer 1

We have made significant improvements of the paper in accordance with your comments. All the corrections are indicated in a highlighted changes file.

Reviewer 2 Report

The popularity and application of mobile robotic systems are growing rapidly in today’s world.At the same time, problems related to the safety of a group of mobile robots are created and actively studied as an interesting and hot topic.This article is based on the method for detecting abnormal activity in a group of mobile robots and creates an integrated system for detecting anomalies in robotic systems. A set of universal analyzed metrics which characterizes the static behavior of a robotic system for the further development of an anomaly detection system is defined and the technique proposed in this study differing from the exiting ones allows the node to analyze its own behavior dependently.The experimental study show the validity of the method.As the above mentioned,I am favorable to the acceptance of the work before it can be up to the standard published on the journal.Please pay particular attention to English grammar, spelling, tense and sentence structure, especially in Abstract and Conclusions.

Author Response

Reviewer 2

Comments and Suggestions for Authors

The popularity and application of mobile robotic systems are growing rapidly in today’s world. At the same time, problems related to the safety of a group of mobile robots are created and actively studied as an interesting and hot topic. This article is based on the method for detecting abnormal activity in a group of mobile robots and creates an integrated system for detecting anomalies in robotic systems. A set of universal analyzed metrics which characterizes the static behavior of a robotic system for the further development of an anomaly detection system is defined and the technique proposed in this study differing from the exiting ones allows the node to analyze its own behavior dependently. The experimental study show the validity of the method. As the above mentioned, I am favorable to the acceptance of the work before it can be up to the standard published on the journal. Please pay particular attention to English grammar, spelling, tense and sentence structure, especially in Abstract and Conclusions.

Response to Reviewer 2

Dear Reviewer,

Thank you very much for your feedback. We have paid great attention to the article design in accordance with the journal requirements. Also, we have improved English usage in the manuscript as much as possible within the allotted time of 10 days. All the changes can be seen in the revised manuscript.

Round 2

Reviewer 1 Report

The Authors made thorough work. The paper is now polished and in my opinion can be accepted.

This manuscript is a resubmission of an earlier submission. The following is a list of the peer review reports and author responses from that submission.

Round 1

Reviewer 1 Report

Dear Authors,

In this paper, you focus on possible cyber security attacks on a group of mobile robots.

This paper presents an experimental work, and it is clear that you investigated many communication protocols and attacks. That said, the overall merit of the paper and the significance of the contributions are rather questionable.

To begin with, the paper needs significant language editing. Many sentence are completely meaningless. Moreover, your paper includes some very questionable statements, without any citations and references. For example, you write:

"According to a recently conducted by Transparency Market Research study of mobile robotics market is projected that by 2019 the annual rate of growth of the world market of mobile robots will rise to 12.6%."

Where is the origin of this study? 12.6% of what?

Similarly, you state:

"If the attacker's task is to penetrate the network, gain access to all devices on the network, disrupt the logic of the system, and destabilize the network, and receive confidential information, then the attacker will implement internal active attacks on the network..."

What is the origin of this observation? Is that an assumption that you're making in this paper?

On a similar note, your literature review (and the cited references) are very limited, and focused on a very small group of authors (many self-citations), and does not even begin to explore the space of the current-state-of-art. Neither in the space of mobile robots, nor in the space of security for robotic system and cyber-physical systems. 

Moreover, you never actually present the whole experimental setup, leaving the whole experimental analysis completely unclear. All of the sudden, you are using some security and pen testing tools, without explaining why, and without explaining the tool itself and its purpose.  

You listed the used commands used, and that's great, but what are you actually trying to test by running those commands?

For all of these reasons, and more, the paper does not seem to be ready in its current form.

Author Response

Thank you for yor revision. I have tried to consider all your comments in the correct article.I tried to more clearly explain the results of the experiments, provide additional explanatory results.Removed information related to the implementation details of the experiments. Unfortunately, it is difficult to discharge stae cover fully all aspects of an experimental stand and scripting attacks, is more engineering solutions, so I dropped them.References has been expanded.

Reviewer 2 Report

The English needs attention and the grammar and syntax is often a bit strange.

A native English speaker should revise the text for incorrect tenses and for incorrect sentence syntax throughout the paper. Examples

Line 26

Line 32

Line 35

Line 36

Line 45 etc

Table 2 has non-English characters in it. Same at Line 70

Line 115 – 126 has a formatting issue. Either use bullet points for all or for none, but not a mixture.

At Line 150 – the wrong figure is referenced in the text.

The titles “Figure 1” to  “Figure 4” are used twice each for different diagrams.

The discussion of the first Figure 2 does not clearly indicate which node is the central server. Is this the test bed that is used for the experiments?

The acronym “CBR” is not explained, and this may not be obvious to all readers. I am assuming it refers to “constant bit rate”?

In section 4 the IP address used in the explanation of the settings for Metasploit does not match the IP address of any node in the diagram of the network from figure 2.

It is also not clear in the discussion which are the “attacker” nodes and which are the “trusted” nodes or how many of each there are during the experiments. The discussion then goes on to talk about 10 nodes.

The use of the TCP protocol is a bit confusing, as the authors’ discussion in section 3 states that the network uses UDP to transfer data. Or are they using TCP to block the UDP packets? This needs to be clarified.

The explanation of the black hole/grey hole (starting at Line 225) needs to be clarified, as the difference between these two attacks is not clearly articulated for the reader.

At Line 233 the text should make it clear that the client will receive the message “com2” only if the attack is successful.

At Line 236 the authors introduce “habu” and “nfqsed” tools without explaining that these are a penetration tool and a tool for transparently modifying network traffic. The discussion of the attack using  iptables does not appear from the text to send out a packet containing “com2”, as all packets are dropped. Or are these two different experiments?

At Line 248 – the tool mdk3 is introduced, again without explaining that this exploits IEEE 802.11 protocol weaknesses.

At Line 391 the discussion is about Figure 3 but the text references Figure 4 – this is an error.

Line 393 – a formula is given and this needs more clarification

At Line 421 the figure numbering changes back to “Figure 1” although the text references “Figure 6”. Similar for “Figure 7”, “Figure 8” and “Figure 9”.

The style of the report needs to be tidied up as it is a bit muddled at the moment. The experimental setup needs to be discussed.

Line 509 – the table has non-English words present.

The experiments need to be properly described, as it is not clear to the reader what they have actually done. The different numbers of robots/nodes and different numbers of attacking nodes need to be explained for the different experiments.  How many experiments have been completed?

The results – line 417 – the discussion of figure 6 (wrongly labelled) is very sparse and needs more explanation.

The experiments performed should be explained separately and in detail. Currently they are very difficult to follow and the discussion is disjointed.

Author Response

Many thanks for your review. I tried to take into account your comments about the quality of the English language. Corrections related to defects in the article were made. The section related to the experimental data has been significantly reworked. The description of the experimental stand was modified taking into account the comments.

Round 2

Reviewer 1 Report

Dear authors,

While you have made improvements to your manuscript, in accordance with the reviewers' comments, your paper still exhibits significant lack of knowledge of the existing related work. 

 Topics of interest include:

-security and privacy issues related to wireless, ad hoc and wireless sensor networks

- security and privacy issues related cyber-physical systems

-security and privacy issues related to robotic and autonomous systems

- experimental analysis of attacks against robotics systems

More importantly, your paper starts out as investigating attacks against mobile robotic systems, but your experimental work quickly becomes an investigation of a know problem of denial-of-service attacks against mobile wireless networks. This is a well-investigated research area, and you may want to check some of the existing work, and investigate how already known results apply to mobile robotic systems.

Author Response

Good day. I expanded the second section of the article, added a review of studies on related subjects at the DOS attacks, and added the conclusions of the existing problems in the development of intrusion detection systems.

Reviewer 2 Report

The paper is much improved. However, at lines 231 and 242 the section has the same number (4), which makes the following sections wrongly numbered. There are also still a few places where the standard of English could be improved.

Author Response

Good day. I changed the recurring numbering. And tried to improve English in critical places.

Round 3

Reviewer 1 Report

Dear authors,

Thank you for improving your background and related work section. It still does not give an overview of all of the related and relevant work, but it is considerably improved, compared to earlier versions.

That said, the major concern about your paper still remains - what is the contribution of your paper. 

How does one take your experimental analysis and infer meaningful information from it? How does one take your proposed metrics, and apply them to other mobile robotic systems?

Most importantly, how is your experimental research different than a well-known related research in the area of ad hoc, mesh and wireless sensor networks?